# A Comparison of Equation Córdoba for Estimation of Body Fat (ECORE-BF) with Other Prediction Equations

**DOI:** 10.3390/ijerph17217940

**Published:** 2020-10-29

**Authors:** Rafael Molina-Luque, Aina M Yañez, Miquel Bennasar-Veny, Manuel Romero-Saldaña, Guillermo Molina-Recio, Ángel-Arturo López-González

**Affiliations:** 1Department of Nursing, Pharmacology and Physiotherapy, School of Medicine and Nursing, University of Córdoba, 14004 Córdoba, Spain; rafael.moluq@gmail.com (R.M.-L.); romero@enfermeriadeltrabajo.com (M.R.-S.); gmrsurf75@gmail.com (G.M.-R.); 2Nursing and Physiotherapy Department, University of the Balearic Islands, 07122 Palma, Spain; miquel.bennasar@uib.es; 3Research Group on Evidence, Lifestyles & Health, Health Research Institute of the Balearic Islands (IdISBa), 07010 Palma, Spain; angarturo@gmail.com; 4Prevention of Occupational Risk in Health Services, Balearic Islands Health Service, 07003 Palma, Spain; 5University School of Odontology ADEMA, University of the Balearic Islands, 07009 Palma, Spain

**Keywords:** adults, anthropometry, body fat, obesity

## Abstract

There are multiple formulas for estimating the percentage of body fat (BF%). Clínica Universidad de Navarra-Body Adiposity Estimator (CUN-BAE) is one of the most used formulas because of its accuracy and its association with cardiovascular pathologies. Equation Córdoba for Estimation of Body Fat (ECORE-BF) was developed to simplify the calculation of BF% while maintaining a similar level of accuracy. The objective was to compare ECORE-BF in a large sample of Spanish workers using CUN-BAE as a reference. A cross-sectional study was carried out on 196,844 participants. The BF% was estimated using different formulas: relative fat mass (RFM), Palafolls, Deurenberg, and ECORE-BF. The accuracy of the estimation was determined using Lin’s concordance correlation coefficient (CCC) and the Bland–Altman method, using CUN-BAE as the reference method. ECORE-BF reached the highest concordance (CCC = 0.998). It also showed the lowest mean difference (−0.0077) and the tightest agreement limits (−0.9723, 0.9569) in the Bland–Altman test. In both analyses, it remained robust even when separating the analyses by sex, nutritional status, or age. ECORE-BF presented as the most straightforward and most accurate equation for the estimation of BF%, remaining robust regardless of population characteristics.

## 1. Introduction

Overweight and obesity, defined as the accumulation of fatty tissue, are global public health problems [1,2] and their prevalence has increased steadily in recent years [3]. This epidemic has made chronic diseases increasingly common worldwide, showing high rates in clinical and community settings. An abnormal amount of fat tissue is a risk factor for type 2 diabetes mellitus, hypertension, atherogenic dyslipidemia, and cardiovascular disease, among others [4,5,6].

Body Mass Index (BMI) has been traditionally used for the assessment of nutritional status. However, its use in isolation is controversial because it does not address the distribution of body compartments. Moreover, it does not consider the influence of other variables such as age or sex, which are fundamental in estimating body composition [7]. These issues cause it to have a low correlation and concordance with body fat (BF) [8,9]. Therefore, the same BMI value can be associated with a wide range of body fat percentages, from healthy to pathological, making clinical assessment difficult [10].

Many tools are available to obtain BF accurately, such as dual-energy X-ray absorptiometry (DXA), magnetic resonance imaging (MRI), and air displacement plethysmography (ADP), among others [11]. However, their large size (which makes them difficult to transport), high cost, complexity, and the use of radiation, among other problems, make them challenging to use in many clinical and community settings [8,12]. Different formulas and other indices from anthropometric variables have been proposed in various populations to avoid these problems, and they have shown a higher correlation with BF than BMI [13,14,15]. Additionally, these equations are simple to use, accessible, and have shown a strong association with various chronic diseases [16].

Among these different equations, the Clínica Universidad de Navarra-Body Adiposity Estimator (CUN-BAE) stands out. CUN-BAE was validated using ADP as a gold standard, and it is widely used in clinical studies to discriminate against various cardiovascular pathologies [16,17]. The complexity of the formula, composed of nine components, makes it challenging to use in real clinical settings where the appropriate software is not available. Several authors have developed alternatives to the equation in order to respect Ockam’s razor principle and facilitate accessibility in any condition [15,18]. In our group, the Equation Córdoba for Estimation of Body Fat (ECORE-BF) was developed [15]. It shows a very high concordance (0.960) with CUN-BAE and requires only three components for its calculation. However, the equation has not been tested in a large sample with different characteristics from those used for its development.

Therefore, the objective of the study is to compare ECORE-BF with different BF prediction equations in a large sample of Spanish workers.

## 2. Materials and Methods

### 2.1. Study Design, Population, and Sample

A cross-sectional study was carried out on the working population of the Balearic Islands (Spain) belonging to different productive sectors (public administration, services industries, healthcare, or postal office). The sample was selected from the workers who carried out health surveillance in the period from 2012 to 2016. Prior to each occupational health examination day, workers were assigned a number, and half of them were selected using a random number table. Workers who had been selected were invited to participate in the study during the occupational health examination. The following inclusion criteria were considered: aged 18 to 65 years, agreement to participate in the study, and gainfully employed. A total of 196,844 workers participated in the study sample.

### 2.2. Study Variables and Measurement

The body fat percentage (BF%) was estimated through different formulas that have calculations based on BMI, waist circumference (WC), or both. The formulas used were CUN-BAE (gold standard) [17], an equation formed by nine components and that uses age, BMI, and sex for its calculation; ECORE-BF [15], which works with natural logarithm of body mass index (lnBMI), age, and sex; relative fat mass (RFM) [13], which includes height, WC, and sex; Palafolls [14], which uses BMI, WC, and sex; and Deurenberg [19], which utilizes BMI, sex, and age. The formulas used are specified in Table 1.

In addition, age (years), sex (female/male), height (cm), weight (kg), BMI (kg/m^2^), and WC (cm) were collected. Nutritional status was classified according to the WHO criteria for BMI [20]. CUN-BAE and cut-off points for Caucasians were used for BF% (for women: normal weight, <30%; overweight, 30.1–35%; obese, >35.1%; and for men: normal weight, <20%; overweight, 20.1–25%; obese, >25.1%) [21].

Anthropometric variables were measured according to the recommendations of the International Standards for Anthropometric Assessment of International Society for the Advancement of Kinanthropometry (ISAK) [22]. The weight was measured with an electronic scale (Seca 700 scale, Seca GmbH, Hambrug) with an accuracy of 0.1 kg. The height was measured with a stadiometer (Seca 220 (CM) telescopic height rod for column scales, Seca GmbH, Hamburg) with an accuracy of 0.5 cm. The WC was measured at the midpoint between the last rib and the iliac crest. All measurements were made by trained staff to minimize the coefficient of variation. Each measurement was made three times, and the average was registered.

### 2.3. Ethical and Legal Aspects

All of the workers were informed, verbally and in writing, of the research objectives before being enrolled. Informed consent was obtained from each in compliance with the current regulations. The study’s protocol complied with the Declaration of Helsinki for conducting medical research involving human subjects and was approved by the Balearic Ethical Committee of Clinical Research (CEI-IB Ref. No:1887). No patients were involved in data analysis or manuscript writing, and the results of the research will not be disseminated to the patients.

### 2.4. Statistical Analysis

The quantitative variables are presented as the mean and standard deviation, and the qualitative values displayed as absolute and relative frequencies.

To contrast the goodness-of-fit to a normal distribution of the data from the quantitative variables, the Kolmogorov–Smirnov test with the Lilliefors correction was employed. The Student’s t-test for two means and ANOVA for three or more means was performed for the bivariate hypothesis contrast. For the correlation between the quantitative variables, the Pearson’s correlation coefficient (r) was used.

We also analyzed the degree of concordance with the reference method quantitatively, computing the Lin’s concordance correlation coefficient (CCC), and graphically, with the Bland–Altman method. Correlation and concordance analyses were computed according to sex, nutritional status (BMI), and quartiles of age.

For all of the statistical analyses, the probability of an α error of below 5% (p < 0.05) was considered statistically significant. Confidence intervals were calculated at 95%. For the statistical analysis, IBM SPSS Statistics 22.0 software (IBM, Chicago, IL, USA) and Epidat 4.2. (Department of Sanidade, Xunta de Galicia, Galicia, Spain) were used.

## 3. Results

Of the 196,844 workers, 41.7% were women. The overall mean age was 40 (SD 10.6) years. The BF% was 29.5% (SD 8.2) and higher in women (*p* < 0.001). Following the BMI criteria, 36.1% (95% CI: 35.9–36.3) and 17.8% (95% CI: 17.6–18) of the subjects were overweight and obese, respectively. All variables analyzed showed statistically significant differences between men and women (*p* < 0.001) (Table 2).

### 3.1. Comparison of the Different Estimation Formulas and Adiposity Indexes

#### 3.1.1. Correlations between the Different Formulas and CUN-BAE

ECORE-BF showed the highest correlation values with CUN-BAE in general, in women, and in men, followed by Palafolls and Deurenberg. In the case of these latter formulas, the correlation decreased when the analysis was separated by sex. Additionally, the analysis of linear correlations according to the BMI categories and age showed that ECORE-BF was the most robust equation. Palafolls and Deurenberg showed similar values, although lower than that of ECORE-BF. Finally, RFM was the equation with the lowest correlation values, especially when separating the analysis according to sex and in participants with a healthy weight (Table 3).

#### 3.1.2. Lin’s Concordance between the Different Formulas and CUN-BAE

ECORE-BF achieved the highest concordance, remaining robust after separating the analysis between men and women, BMI categories, and age. Something similar was observed in the case of Deurenberg, although the CCC values were lower. Palafolls showed a concordance of 0.836 in the total sample. However, when the analysis was separated according to sex, BMI categories, and age, the CCC decreased in most groups, especially in men, participants with a normal weight, and those under 32 years old. The CCC increased in the case of obesity status and those over 49 years old. Finally, RFM showed the lowest concordance values, a tendency highlighted by a CCC of 0.527 for the obesity group (Table 4).

The Bland–Altman graphics (Figure 1) showed that ECORE-BF was the prediction equation with the smallest difference in means and the tightest agreement limits (Table 5). In other words, the estimations with ECORE-BF were those that showed the smallest dispersion when compared with those of CUN-BAE; moreover, the results remained consistent in men, women, all BMI categories, and all age groups. The rest of the equations studied showed higher mean differences, as well as more open limits of agreement, evidenced by a higher dispersion according to CUN-BAE.

## 4. Discussion

Our study indicates that ECORE-BF shows a high concordance with CUN-BAE, a widely used method to estimate body fat percentage in the general population. The estimation and distribution of body fat have been widely studied, mainly because of its involvement in the development of various chronic diseases [23]. In this context, several formulas based on anthropometric measurements have been developed to replace more complicated and expensive tools [13,14,15].

ECORE-BF is shown to be an accessible and easy to calculate alternative that has a high predictive capacity [15]. Our results show an excellent clinical concordance with CUN-BAE, reaching a value of 0.998 in the total sample, which is higher than that observed in the original study (Intraclass Correlation Coefficient= 0.960). This concordance remained stable in all performed analyses, which confirms its robustness regardless of the population where it is applied. Furthermore, a mean difference close to 0 was found, with very close agreement limits (Low Limit of Agreement (LLA) = −0.9723, Upper Limit of Agreement (ULA) = 0.9569), improving the results found in the development of the equation (Mean Difference (MD) = 1.819; LLA = −2.256, ULA = 5.895) [15]. It should also be noted that of the rest of the formulas studied, the one proposed by Deurenberg et al. was the only one that showed a similar agreement to ECORE-BF (CCC = 0.981), although the dispersion in the estimate was higher (MD = 0.70; LLA = −3.42, ULA = 2.02) than ECORE-BF.

Although BMI alone correlates significantly with BF%, it has a low predictive capacity [7,24,25]. However, when other variables, such as sex and age, are incorporated into the analysis, the accuracy improves [12,15,17]. Hastuti et al. developed several formulas using anthropometric measurements. They showed that equations using BMI and sex produced good results in explaining variability (R^2^ = 0.668) and in the limits of agreement (−0.2 ± 4.3 in men and 0.4 ± 5.4 in women) [24]. The rest of the formulas developed by these authors presented similar precision results. However, for their calculation, they required variables that were difficult to obtain, such as skin folds [24].

Another variable used for the estimation of the BF% is WC. When WC is included in the regression models, it achieves positive results in explained variability and concordance but maintains a high dispersion in the estimates [26]. Palafolls and RFM are two formulas that incorporate WC into their analyses. Palafolls, which uses BMI, sex, and WC, was developed following the same methodology as that used for ECORE-BF, but in a smaller sample [14]. Our results show that the CCC was 0.836, although the dispersion in the estimate was higher than that shown by ECORE-BF (MD = 3.88; LLA = 0.66, UPA = 8.43). The dispersion decreased in three of the four age groups in which the analysis was separated. Furthermore, RFM, developed using the DXA as a reference, showed the worst results of the equations studied. Although the linear correlation with CUN-BAE was good (r = 0.821), RFM showed the lowest degree of agreement (CCC = 0.765), results that align with those shown by other authors. Paek et al. studied the validity of RFM in a sample of Korean adults; the results showed that it had a high dispersion in the estimate, with a tendency to overestimate, and a predictive capacity similar to that shown by the BMI [27]. Although Guzmán-León et al. stated that RFM presents greater consistency in the estimation of BF% than BMI, they did not carry out a study of clinical, quantitative, or graphic concordance, and conclusions had been drawn from the estimates obtained through a new linear regression model. In other words, to determine the accuracy of RFM in their sample, they carried out a new regression model with the equation, which implies an adjustment with new beta coefficients. This calculation guarantees that the formula explains a higher variability than what would be achieved without the original equation’s adjustment [28]. Additionally, our results show that the degree of agreement with CUN-BAE was higher in participants with normal weight and overweight (CCC = 0.741 and CCC = 0.767, respectively), dropping more than 0.2 points in those with obesity (CCC = 0.527). This decrease is because the BF% estimate is made using a central obesity indicator (WC), which means that it is not entirely accurate in people with a high overall BF%. Other developed equations that use WC as the adjustment variable also introduce weight, ensuring that a measure is available that accounts for total body mass [29].

Several authors have developed formulas for the estimation of BF% composed of a large number of variables. However, the results obtained do not improve on those achieved by simpler models, such as ECORE-BF [30]. Therefore, making the equations more complex does not guarantee greater precision in the estimation. Accessible models should be developed that help clinical staff diagnose overweight and obesity easily [31]. As mentioned above, equations based on anthropometric measurements (e.g., BMI) have shown good results [15,17,18,26]. However, these equations generally have limitations and are less accurate than those based on skin folds [32]. Nevertheless, they are widely used in the scientific literature and are good indicators of cardiovascular health [16,33,34]. ECORE-BF is, therefore, an alternative that can be used in specific clinical settings where qualified personnel are not available to take complex anthropometric measurements (e.g., skin folds). In short, all tools (DXA, ADP, equations based on anthropometry) may have some limitations [35,36], but knowing them and identifying which situations to use them in can be useful when interpreting the results they provide.

## 5. Study Limitations and Strengths

The main limitation of the study is that a gold standard, such as DXA or ADP, was not used. Future studies should validate ECORE-BF by using more accurate methods. However, ECORE-BF has demonstrated robustness in working populations regardless of characteristics and sample size. Finally, it is important to emphasize that ECORE-BF is validated only in the working Caucasian population. Therefore, in order to extend its use to populations with other characteristics, it must be validated in those populations or adapted to them, as has been done previously with other equations [37].

## 6. Conclusions

ECORE-BF has only three components, which makes it easy to calculate, regardless of the clinical setting in which it is used. Furthermore, the simplicity of obtaining the variables involved in the estimation reduces the need for highly qualified personnel trained in anthropometric measurement. Finally, the robustness found in men, women, and any age group or nutritional status guarantees the accuracy of the results in any worker to whom it is applied.

## Figures and Tables

**Figure 1 ijerph-17-07940-f001:**
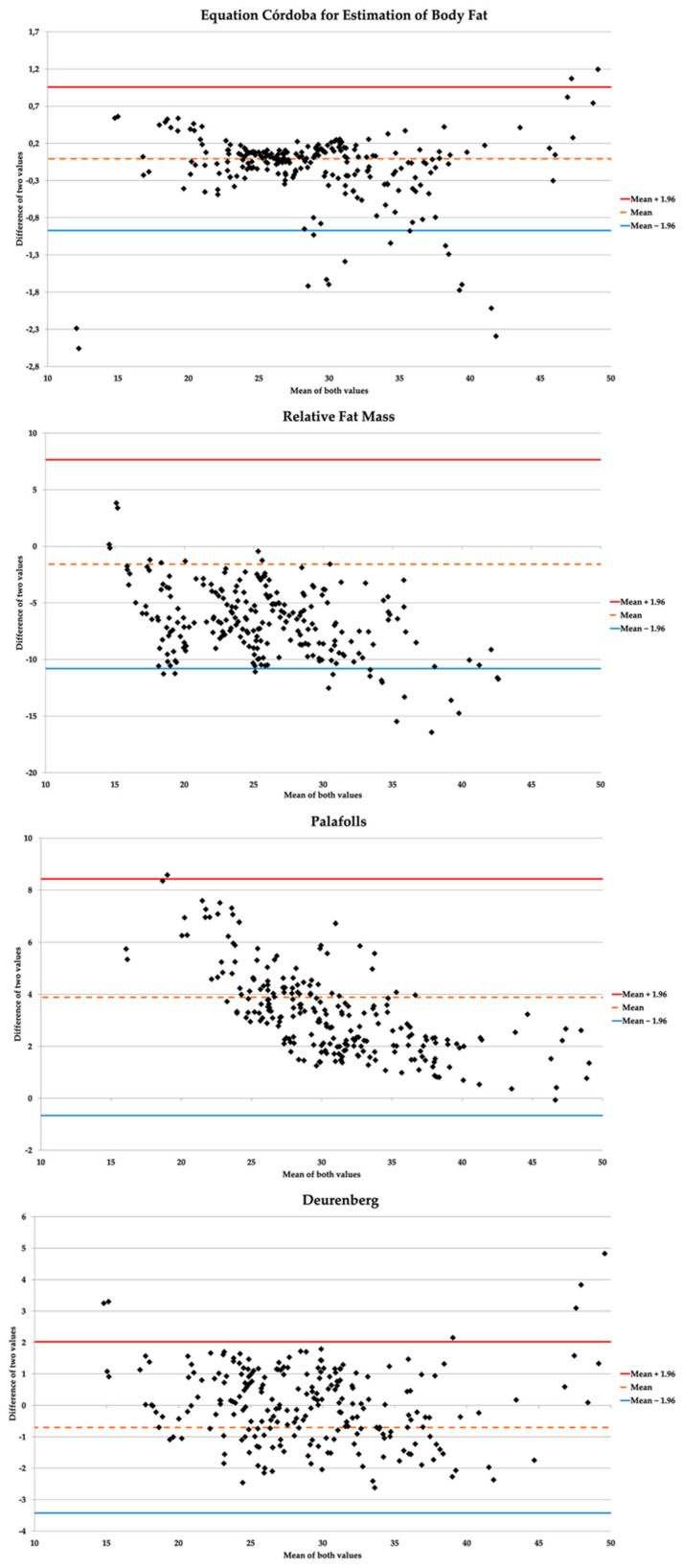
Concordance graphs between study formulas and CUN-BAE (*gold standard)*.

**Table 1 ijerph-17-07940-t001:** Estimation formulas.

Name	Formula
CUN-BAE (BF%)	−44.988 + (0.503 × Age) + (10.689 × Sex) + (3.172 × BMI) − (0.026 × BMI^2^) + (0.181 × BMI × Sex) − (0.02 × BMI × Age) − (0.005 × BMI^2^ × Sex) + (0.00021 × BMI^2^ × Age)
ECORE-BF (BF%)	−97.102 + (0.123 × Age) + (11.900 × Sex) + (35.959 × LnBMI)
RFM (BF%)	64 − (20 × [height/WC]) + (12 × Sex)
Palafolls (BF%)	([BMI/WC] × 10) + BMI + (10 × Sex)
Deurenberg (BF%)	(1.20 × BMI) + (0.23 × Age) − (10.8 × Sex) − 5.4

CUN-BAE: Clínica Universidad de Navarra-Body Adiposity Estimator; ECORE-BF: Equation Córdoba for Estimation of Body Fat; LnBMI: natural logarithm of body mass index; RFM: relative fat mass; BMI: body mass index; WC: waist circumference. CUN-BAE, ECORE-BF, RFM, and Palafolls: men = 0 and women = 1; Deurenberg: men = 1 and women = 0.

**Table 2 ijerph-17-07940-t002:** Descriptive characteristics of the sample.

Variables	Total (*n* = 196,844)	Women (*n* = 82,104)	Men (*n* = 114,740)	*p*-Value
	Mean (SD) or n (%)			
**Age (years)**	40 (10.6)	39.4 (10.5)	40.4 (10.7)	<0.001
**Weight (kg)**	74.8 (16)	65.8 (13.6)	81.3 (14.3)	<0.001
**Height (cm)**	169.1 (9.3)	161.7 (6.5)	174.4 (7)	<0.001
**WC (cm)**	83.4 (10.7)	75.9 (8.2)	88.7 (8.9)	<0.001
**BMI (kg/m^2^)**	26.1 (4.7)	25.2 (5)	26.7 (4.3)	<0.001
Normal weight	90,733 (46.1%)	47,175 (57.5%)	43,558 (38%)	<0.001
Overweight	71,109 (36.1%)	22,185 (27%)	48,924 (42.6%)	<0.001
Obesity	35,002 (17.8%)	12,744 (15.5%)	22,258 (19.4%)	<0.001
**CUN-BAE (%)**	29.5 (8.2)	35 (7)	25.5 (6.4)	<0.001
Normal weight	41,879 (21.3%)	21,121 (25.7%)	21,595 (18.8%)	<0.001
Overweight	57,765 (29.3%)	22,704 (27.7%)	33,390 (29.1%)	<0.001
Obesity	97,200 (49.4%)	38,279 (46.6%)	59,755 (51.1%)	<0.001
**ECORE-BF (BF%)**	29.5 (8.1)	35 (7.2)	25.5 (6.1)	<0.001
**RFM (BF%)**	27.9 (6)	32.9 (4.5)	24.3 (4)	<0.001
**Palafolls (BF%)**	33.4 (6.6)	38.5 (5.4)	29.7 (4.6)	<0.001
**Deurenberg (BF%)**	28.8 (7.9)	33.9 (7)	25.1 (6.3)	<0.001

WC: waist circumference; BMI: body mass index; CUN-BAE: Clínica Universidad de Navarra-Body Adiposity Estimator; ECORE-BF: Equation Córdoba for Estimation Body Fat; RFM: relative fat mass.

**Table 3 ijerph-17-07940-t003:** Bivariate correlations between study formulas and CUN-BAE.

	ECORE-BF	RFM	Palafolls	Deurenberg
**Total**	0.998	0.821	0.973	0.986
**Women**	0.998	0.730	0.963	0.981
**Men**	0.998	0.710	0.970	0.979
**Normal weight**	0.998	0.756	0.958	0.988
**Overweight**	1	0.831	0.979	0.976
**Obesity**	0.995	0.835	0.987	0.967
**<32 years ***	0.998	0.824	0.985	0.991
**33–40 years ***	0.999	0.834	0.990	0.994
**41–48 years ***	0.999	0.837	0.993	0.995
**> 49 years ***	0.998	0.851	0.993	0.992

CUN-BAE: Clínica Universidad de Navarra-Body Adiposity Estimator; RFM: relative fat mass; ECORE-BF: Equation Córdoba for Estimation Body Fat. * The age groups were computed according to the quartiles of the sample. All correlations showed a p value below 0.001.

**Table 4 ijerph-17-07940-t004:** Lin’s concordance correlation coefficient between study formulas and CUN-BAE.

	ECORE-BF	RFM	Palafolls	Deurenberg
**Total**	0.998	0.765	0.836	0.981
**Women**	0.997	0.625	0.808	0.968
**Men**	0.997	0.662	0.718	0.977
**Normal weight**	0.997	0.741	0.653	0.983
**Overweight**	1	0.767	0.844	0.960
**Obesity**	0.993	0.527	0.950	0.954
**<32 years ***	0.997	0.772	0.715	0.961
**33–40 years ***	0.999	0.794	0.828	0.979
**41–48 years ***	0.999	0.759	0.898	0.990
**>49 years ***	0.998	0.702	0.958	0.989

CUN-BAE: Clínica Universidad de Navarra-Body Adiposity Estimator; ECORE-BF: Equation Córdoba for Estimation Body Fat. * The age groups were computed according to the quartiles of the sample. All correlations showed a *p* value below 0.001.

**Table 5 ijerph-17-07940-t005:** Mean differences between formulas and CUN-BAE.

	Mean Difference (SD)	Limit of Agreement (Lower, Upper)
**ECORE−BF:**		
Total	−0.0077 (0.4922)	−0.9723, 0.9569
Women	−0.022 (0.5037)	−1.009, 0.9652
Men	0.0026 (0.4835)	−0.945, 0.9502
Normal weight	0.0811 (0.4888)	−0.8769, 1.039
Overweight	−0.0151 (0.17)	−0.3492, 0.319
Obesity	−0.223 (0.786)	−1.76, 1.32
<32 years *	0.18 (0.61)	−1.01, 1.37
33–40 years *	−0.07 (0.38)	−0.82, 0.67
41–48 years *	−0.12 (0.35)	−0.8, 0.55
>49 years *	−0.03 (0.52)	−1.05, 0.98
**RFM:**		
Total	−1.57 (4.71)	−10.80, 7.66
Women	−2.04 (4.86)	−11.56, 7.48
Men	−1.23 (4.57)	−10.19, 7.72
Normal weight	0.8 (4.35)	−7.72, 9.32
Overweight	−2.16 (3.32)	−8.68, 4.35
Obesity	−6.51 (3.68)	−13.72, 0.7
<32 years *	1.20 (4.79)	−8.18, 10.57
33–40 years *	−1.23 (4.3)	−9.68, 7.21
41–48 years *	−2.63 (4.1)	−10.63, 5.37
>49 years *	−4.05 (3.84)	−11.56, 3.47
**Palafolls:**		
Total	3.88 (2.32)	−0.66, 8.43
Women	3.48 (2.32)	−1.08, 8.04
Men	4.17 (2.27)	−0.28, 8.63
Normal weight	5.39 (2.23)	−1.02, 9.77
Overweight	2.96 (1.46)	0.1, 5.82
Obesity	1.84 (1.1)	−0.31, 4
<32 years *	6.18 (2.17)	1.93, 10.42
33–40 years *	4.28 (1.6)	1.15, 7.4
41–48 years *	3.02 (1.3)	−0.47, 5.57
>49 years *	1.69 (1.13)	−0.51, 3.9
**Deurenberg:**		
Total	−0.70 (1.39)	−3.42, 2.02
Women	−1.13 (1.37)	−3.82, 1.55
Men	−0.4 (1.31)	−2.98, 2.2
Normal weight	−0.33 (1.1)	−2.49, 1.83
Overweight	−1.06 (1.3)	−3.6, 1.48
Obesity	−0.95 (1.9)	−4.68, 2.78
<32 years *	−1.61 (1.5)	−4.55, 1.34
33–40 years *	−1.11 (1.08)	−3.23, 1.01
41–48 years *	−0.49 (0.87)	−2.19, 1.22
>49 years *	0.56 (0.9)	−1.2, 2.33

CUN-BAE: Clínica Universidad de Navarra-Body Adiposity Estimator; ECORE-BF: Equation Córdoba for Estimation Body Fat. * The age groups were computed according to the quartiles of the sample.

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
