# Peer review of "A Comparison of Equation Córdoba for Estimation of Body Fat (ECORE-BF) with Other Prediction Equations"

_ijerph, 2020, doi:10.3390/ijerph17217940_

Round 1

Reviewer 1 Report

Thank you for the opportunity to read this very interesting paper. The paper has many strengths that should be of interest to the journal audience. Thus, the following suggestions are around enhancing the presentation for publication and clarifying aspects of the data and reporting.

I will go by line number for the most part. If not, I will try to be as specific as possible in noting the area I am speaking about.

Abstract

Which is the meaning of CUN-BAE and ECORE-BF? Acronym?

Population: Spanish workers? The workers who carried out health surveillance?

…the working population of the Balearic Islands (Spain)? Authors must specify it.

Introduction

[Line 37]  [4 – 6] à [4–6] (no spaces). Authors must review it.

Materials and methods

In terms of methodology, this section needs to present a better rationale for the study and the methodology employed. Please, clearly explain what you have done. Also, neither appear information related with inclusion and exclusion criteria, dates, protocol. Authors must specify it.

How was the sample chosen? Authors must specify it.

How have they contacted with the sample?

The formulas used should be include as table or as figure.

[Line 97] … was approved by the Bioethics Committee. Which one? We need ID.

Results and conclusions are correct.

References

According references format, Volume. Author 1, A.B.; Author 2, C.D. Title of the article. Abbreviated Journal Name Year, Volume, page range.

Author Response

Response to Reviewer 1 Comments

Thank you for the opportunity to read this very interesting paper. The paper has many strengths that should be of interest to the journal audience. Thus, the following suggestions are around enhancing the presentation for publication and clarifying aspects of the data and reporting. I will go by line number for the most part. If not, I will try to be as specific as possible in noting the area I am speaking about.

R: Thank you for your kind words. The authors are honoured to receive such a satisfactory evaluation of their work. We will then respond to your suggestions.

Point 1: Abstract: Which is the meaning of CUN-BAE and ECORE-BF? Acronym?

Response 1: Thank you for your appreciation. The words have been described before using their acronyms in the abstract (Lines 19-21).

Point 2: Abstract: Population: Spanish workers? The workers who carried out health surveillance? … the working population of the Balearic Islands (Spain)? Authors must specify it.

Response 2: Thank you for your comment. The authors have clarified in the abstract that the sample is made up of Spanish workers (Line 22).

Point 3: [Line 37] [4 – 6] à [4–6] (no spaces). Authors must review it.

Response 3: Thank you for your appreciation. These and other similar mistakes have been corrected (Line 40).

Point 4: In terms of methodology, this section needs to present a better rationale for the study and the methodology employed. Please, clearly explain what you have done. Also, neither appear information related with inclusion and exclusion criteria, dates, protocol. Authors must specify it.

Response 4: Thank you for your comment. The authors have introduced new information to clarify the methodology section (Lines 72-81), and now reads: “A cross-sectional study was carried out on the working population of the Balearic Islands (Spain) belonging to different productive sectors (public administration, services industries, healthcare, or postal office)”.

Point 5: How was the sample chosen? Authors must specify it.

Response 5: Thank you for your appreciation. However, the text specifies that the sample was selected randomly (Lines 74-81), and now reads: “The sample was selected from the workers who carried out health surveillance in the period from 2012 to 2016. Prior to each occupational health examination day, workers were assigned a number, and half of them were selected using a random number table. Workers who had been selected were invited to participate in the study during the occupational health examination. The following inclusion criteria were considered: age 18 and 65 years, agreement to participate in the study and to be gainfully employed. Finally, a total of 196,844 workers participated in the study sample”.

Point 6: How have they contacted with the sample?

Response 6: Thank you for your comment. The sample was selected from workers who attended the health surveillance examination during the years 2012 and 2016 (this health examination is mandatory in Spain). Each of the workers signed a consent form to provide their data for this and other studies.

Point 7: The formulas used should be include as table or as figure.

Response 7: Thank you for your appreciation. The authors have added a table in which the equations used in the paper have been described (Page 3). Due to this, the rest of the tables have been renumbered, having a total of 5.

Point 8: [Line 97] … was approved by the Bioethics Committee. Which one? We need ID.

Response 8: The authors have specified the name of the Bioethics Committee that approved the study (Lines 107-108). Besides, the identification number of the approval certificate has been added, and now reads: “…was approved by the Balearic Ethical Committee of Clinical Research (CEI-IB Ref. No:1887)”.

Reviewer 2 Report

In this cross-sectional study entitled "Validation of Equation Córdoba for Estimation of Body Fat (ECORE-BF)", Dr. Molina-Luque and co-workers consider an equation already validated on a development sample, which however lacks a cross-validation sample. The idea of ​​filling this gap makes the study's aim interesting, however for the reasons listed below I must suggest rejecting this manuscript.

A serious methodological error militates against the publication of this article. The reference method [the Clínica Universidad de Navarra - Body Adiposity Estimator (CUN-BAE)] used to validate this equation on a cross-validation sample is absolutely not acceptable. When validating an equation, a criterion method that guarantees a high degree of accuracy is chosen as a comparison. CUN-BAE is not included among the criterion methods for the fat mass (FM) estimation, while dual-energy X-ray absorptiometry (DXA), underwater weighing (UWW) or air displacement plethysmography (ADP) can be used for this purpose. Furthermore, the gold standard method for assessing body composition at the molecular level is the 4-compartment model (4C), where FM is obtained from UWW and not DXA as the authors mention in the introduction. Having used CUN-BAE as a reference method does not allow me to make further suggestions for improving the overall quality of the manuscript, where the experimental design must be entirely modified by including 4C, UWW, ADP, or DXA as a validation method. In addition to this concern, other gaps are present in the other sections of the manuscript, but it is not constructive to highlight them at this point. Presenting an equation validated against a doubly-indirect method for the FM estimation would not add any value to the field. In fact, there are no publications in the literature that include CUN-BAE as a reference method for assessing FM.

Author Response

.

In this cross-sectional study entitled "Validation of Equation Córdoba for Estimation of Body Fat (ECORE-BF)", Dr. Molina-Luque and co-workers consider an equation already validated on a development sample, which however lacks a cross-validation sample. The idea of ​​filling this gap makes the study's aim interesting, however for the reasons listed below I must suggest rejecting this manuscript.

A serious methodological error militates against the publication of this article. The reference method [the Clínica Universidad de Navarra - Body Adiposity Estimator (CUN-BAE)] used to validate this equation on a cross-validation sample is absolutely not acceptable. When validating an equation, a criterion method that guarantees a high degree of accuracy is chosen as a comparison. CUN-BAE is not included among the criterion methods for the fat mass (FM) estimation, while dual-energy X-ray absorptiometry (DXA), underwater weighing (UWW) or air displacement plethysmography (ADP) can be used for this purpose. Furthermore, the gold standard method for assessing body composition at the molecular level is the 4-compartment model (4C), where FM is obtained from UWW and not DXA as the authors mention in the introduction. Having used CUN-BAE as a reference method does not allow me to make further suggestions for improving the overall quality of the manuscript, where the experimental design must be entirely modified by including 4C, UWW, ADP, or DXA as a validation method. In addition to this concern, other gaps are present in the other sections of the manuscript, but it is not constructive to highlight them at this point. Presenting an equation validated against a doubly-indirect method for the FM estimation would not add any value to the field. In fact, there are no publications in the literature that include CUN-BAE as a reference method for assessing FM.

R: We would like to thank to the reviewer the time dedicated to revising the manuscript. We agree with the reviewer that our study design cannot be considered as a “standard validation study” because we have not used a recognized gold standard (DXA or plethysmography). For this reason, we have changed the title and all the manuscript references to validation to indicate that we have performed a comparison of a new method with a different fat mass estimation equations.

We believe that our study will be of interest to the clinicians because it compares the results of ECORE-BF for estimation the body fat with other equations in a large sample of adult workers (n=196,844). There are multiple formulas for estimating the percentage of BF. CUN-BAE is one of the most used for its accuracy and its association with cardiovascular pathologies. ECORE-BF was developed to simplify calculation maintaining the accuracy.

Reviewer 3 Report

Dear Authors,

The work presented is of interest to the readers of the journal, especially to clinicians and researchers working with large population samples. The work validates an index of body fat measurement, based on simple anthropometric variables and with a low complex calculation that allows a better analysis of the nutritional situation by including an analysis of body composition beyond the weight/height ratio

The objective is well defined and argued, the experimental design is good, the sample size is high and the statistical analysis is of quality. All this makes the results reliable and, moreover, well discussed.

I think the paper is suitable for publication in the journal. I am only indicating some small suggestions for change regarding some terms used in the writing that I think can facilitate the reader's understanding.

Best regards,

Author Response

Response to Reviewer 3 Comments

Dear Authors,

The work presented is of interest to the readers of the journal, especially to clinicians and researchers working with large population samples. The work validates an index of body fat measurement, based on simple anthropometric variables and with a low complex calculation that allows a better analysis of the nutritional situation by including an analysis of body composition beyond the weight/height ratio. The objective is well defined and argued, the experimental design is good, the sample size is high and the statistical analysis is of quality. All this makes the results reliable and, moreover, well discussed. I think the paper is suitable for publication in the journal. I am only indicating some small suggestions for change regarding some terms used in the writing that I think can facilitate the reader's understanding. Best regards.

R: The authors welcome positive feedback on the work. It is an honour that each section of the work is appreciated. Also, we appreciate your suggestions, which we believe improve the quality of the work.

Point 1: What context is that? This sentence is a little general to be the beginning of the introduction, since the reader does not yet have information about the context of the study.

Response 1: We apologize for the mistake. The sentence lost its original meaning during the translation process. The authors have modified the sentence to make it more easily understandable (Lines 38-39): “This epidemic has made chronic diseases increasingly common worldwide, showing high rates in clinical and community settings”.

Point 2: Formulas and indices from what kind of measures? Some explanation should be added here, such as 'from simple anthropometric variables' or 'from more accessible techniques such as bioimpedance or anthropometry'.

Response 2: Thank you for your comment. The authors have introduced one of the sentences suggested in your commentary to clarify the type of measures applied (Lines 52-53): “Different formulas and other indices from anthropometric variables have been proposed…”.

Point 3: Concordance with what? BF measured by DEXA? Please specify.

Response 3: Thank you for your comment. The authors have specified this point in the introduction section, and now reads: “It showed a very high concordance (0.960) with CUN-BAE and requiring only 3 components for its calculation”.

Point 4: I suggest changing the terms used here to: 'normal body fat, overfat and obesity' to avoid confusion with the classification from the BMI

Response 4: Thank you for your appreciation. The authors agree with your comment. However, the accepted classification in the literature is the one we have used in the text (Gómez-Ambrosi, J., Silva, C., Galofré, J. et al. Body mass index classification misses subjects with increased cardiometabolic risk factors related to elevated adiposity. Int J Obes 36, 286–294 (2012). https://doi.org/10.1038/ijo.2011.100). Therefore, we consider that it should be maintained to ensure homogeneity in terms.

Point 5: Which Committee? Is it from the University of Cordoba, of Balearic Islands, other?. Please specify and provide approval reference if exists

Response 5: Thank you for your correction. The authors have specified the name of the Bioethics Committee that approved the study (Lines 107-108). Besides, the identification number of the approval certificate has been added, and now reads: “…was approved by the Balearic Ethical Committee of Clinical Research (CEI-IB Ref. No:1887)”.

Point 6: I suggest to remember here again that it is regarding the CUN-BAE, considered gold standard for this study.

Response 6: Thank you for your comment. The authors have introduced the clarification you have suggested (Lines 138-139), and now reads: “The ECORE-BF showed the highest correlation values in general, in women and men with CUN-BAE”.

Point 7: I suggest 'Bland Altman graphical analysis' rather than 'study'

Response 7: Thank you for your appreciation. The authors have made the suggested change (Lines 160-161), and now reads: “The Bland–Altman graphics (Figure 1) showed that ECORE-BF was the prediction equation with the smallest difference in means and the tightest agreement limits (Table 5)”.

Round 2

Reviewer 2 Report

I thank the authors for answering my comments and admitting the limitations of their study.

However, I am still convinced that this manuscript provides results of scant interest in the academic field and absolutely not innovative for a journal with a high impact factor.

Comparing the results of a predictive equation with others obtained from different formulas is not enough to conduct a high-quality study.

Again, the statistical analysis is carried out flawlessly but CUN-BAE equation cannot represent a criterion method to evaluate the accuracy of your formula in predicting Fat Mass (FM); this represents a serious methodological error.

Author Response

Thank you for your comments, we agree that ideally ECORE-BF should be compared with a gold standard as DEXA, plethysmography, or magnetic resonance. But we are still convinced that our results are interesting because show a comparison between different body fat prediction equations and allow us to evaluate the utility of ECORE-BF in a large sample.